# Vacuum Brazing of Metallized YSZ and Crofer Alloy Using 72Ag-28Cu Filler Foil

**DOI:** 10.3390/ma15030939

**Published:** 2022-01-26

**Authors:** Liang-Wei Huang, Ren-Kae Shiue, Chien-Kuo Liu, Yung-Neng Cheng, Ruey-Yi Lee, Leu-Wen Tsay

**Affiliations:** 1Department of Materials Science and Engineering, National Taiwan University, Taipei 10617, Taiwan; i13501350@iner.gov.tw; 2Nuclear Fuels and Materials Division, Institute of Nuclear Energy Research, Taoyuan 32546, Taiwan; ckliu2@iner.gov.tw (C.-K.L.); yncheng@iner.gov.tw (Y.-N.C.); rylee@iner.gov.tw (R.-Y.L.); 3Department of Optoelectronics and Materials Technology, National Taiwan Ocean University, Keelung 20224, Taiwan; b0186@mail.ntou.edu.tw

**Keywords:** vacuum brazing, metallization, interfacial reaction, gas-tight

## Abstract

The study focused on dissimilar brazing of metallized YSZ (Yttria-Stabilized Zirconia) and Crofer alloy using BAg-8 (72Ag-28Cu, wt%) filler foil. The YSZ substrate was metallized by sequentially sputtering Ti (0.5/1 μm), Cu (1/3 μm), and Ag (1.5/5 μm) layers, and the Crofer substrate was coated with Ag layers with a thickness of 1.5 and 5 μm, respectively. The BAg-8 filler demonstrated excellent wettability on both metallized YSZ and Crofer substrates. The brazed joint primarily consisted of Ag-Cu eutectic. The metallized Ti layer dissolved into the braze melt, and the Ti preferentially reacted with YSZ and Fe from the Crofer substrate. The globular Fe_2_Ti intermetallic compound was observed on the YSZ side of the joint. The interfacial reaction of Ti was increased when the thickness of the metallized Ti layer was increased from 0.5 to 1 μm. Both brazed joints were crack free, and no pressure drop was detected after testing at room temperature for 24 h. In the YSZ/Ti(0.5μ)/Cu(1μ)/Ag(1.5μ)/BAg-8(50μ)/Ag(1.5μ)/Crofer joint tested at 600 °C, the pressure of helium decreased from 2.01 to 1.91 psig. In contrast, the helium pressure of the YSZ/Ti(1μ)/Cu(3μ)/Ag(5μ)/BAg-8(50μ)/Ag(5μ)/Crofer joint slightly decreased from 2.02 to 1.98 psig during the cooling cycle of the test. The greater interfacial reaction between the metallized YSZ and BAg-8 filler due to the thicker metallized Ti layer on the YSZ substrate was responsible for the improved gas-tight performance of the joint.

## 1. Introduction

To form a metallurgical bond between metal and ceramic materials has been a challenging task over the past 20 years [1,2,3,4,5]. Dissimilar vacuum brazing is an alternative approach to obtaining a metal-ceramic joint. With the selection of appropriate active filler metal, the molten braze melt can wet and bond metal and ceramic substrates together [6,7,8,9,10]. As the coefficients of the thermal expansion of metal and ceramic materials are quite different, enormous residual thermal stresses are induced in the brazed joint. The reliability of the joint after brazing thus deteriorates [11,12,13]. Additionally, many traditional braze fillers cannot wet the ceramic surface and achieve a reliable brazed joint [14,15,16,17,18]. Both issues must be considered before brazing metal and ceramic substrates.

The thermal expansion coefficient of Crofer alloy (12.8 ppm/°C) is very close to that of YSZ (Yttria-stabilized zirconia, 11.4 ppm/°C) [19]. Therefore, the residual thermal stress of the brazed joint is minimized. It is believed that the combination of Crofer and YSZ is one of the best choices for metal–ceramic joining due to their close thermal expansion coefficients. The wetting of the YSZ substrate can be achieved by active brazing [20,21]. Active braze alloys, e.g., Ag-Cu-Ti fillers, have been widely studied in brazing metal and ceramic materials [6,22,23]. The active ingredient, Ti, in the braze easily reacts with and wets most ceramic materials, and a reliable bond is obtained. However, the Ti in the active braze alloy also reacts with the metallic substrate during brazing. The formation of interfacial intermetallic phases between the braze alloy and metallic substrate could also decrease the joint reliability [24]. Decreasing the amount of Ti in the active braze alloy can reduce the interfacial reaction between the braze alloy and metallic substrate.

It has been reported that Cu/Ti-rich melt has high wetting with alumina [25]. The introduction of the Ag layer combined with Cu/Ti layers can produce a liquid film with a chemical composition similar to that of the Ag-Cu-Ti active filler melt on the YSZ substrate during the heating cycle of brazing. As the active element, Ti, is mainly confined at the interface between the metallized YSZ and braze filler, the traditional Bag-8 filler foil can be used to dissimilar brazing the metallized YSZ and Crofer. Therefore, a YSZ substrate sequentially coated with Ti, Cu, and Ag layers with the aid of a magnetron sputtering facility is proposed in the study.

As YSZ is a suitable solid-electrolyte material, many applications for YSZ-brazed joints require that the joints be gas-tight. Some such applications are oxygen transport membranes, gas sensors and fuel cells [26,27,28]. This investigation focused on brazing metallized YSZ and Crofer substrates using the BAg-8 braze alloy for gas-tight applications. The microstructural evolution of the brazed joints and correlations with pressure drop tests at 25 and 600 °C were also evaluated in the experiment.

## 2. Materials and Experimental Procedures

Crofer (VDM Metals, Werdohl, Germany) and YSZ (Kceracell, Republic of Korea) substrates were machined into square sheets with an area of 10 × 10 mm^2^ and thickness of 1 mm. The chemical composition of Crofer in wt% is (20–24)Cr, 0.03C, 0.04N, 0.006S, 0.8Mn, (0.1–0.6)Si, 0.1Al, (1–3)W, (0.2–1)Nb, (0.02–0.2)Ti, (0.04–0.2)La, 0.05P, 0.5Ni, 0.5Cu, and Fe balance. BAg-8 foil (72Ag-28Cu in wt%) with a thickness of 50 μm was used to braze metallized YSZ and Crofer substrates. The eutectic temperature of BAg-8 filler foil is 780 °C. Metallization was conducted by magnetron sputtering. YSZ and Crofer substrates were pre-heated at 200 °C under a vacuum of 10^−3^ Pa. Afterwards, the sputtering chamber was purged with argon to achieve a vacuum of approximately 1 Pa, and sputtering was performed at a constant power for 600 s. For investigation of the effects of coating thicknesses on the microstructure and gas-tight properties of YSZ/Crofer-brazed joints, two different coating combinations were compared. Ti (0.5/1 μm), Cu (1/3 μm), and Ag (1.5/5 μm) layers were sequentially deposited on YSZ, and Ag (1.5/5 μm) layers were deposited on Crofer substrates. The sputtering parameters are shown in Table 1. Figure 1 shows the cross-sectional backscattered images (BEIs) and element mappings of two metallized substrates; all deposited layers demonstrated good quality and adhered to the neighbor layer(s) or Crofer/YSZ substrates.

The BAg-8 filler ball with the weight of 0.15 g was placed on the substrate, and subsequently heated at 850 °C for 600 s in vacuum. The wetting angle was measured from the photo of the transverse view of the specimen. BAg-8 foil was used to braze the metallized Crofer and YSZ. Vacuum brazing was performed with a heating rate of 0.17 °C/s and holding at 850 °C for 600 s. Finally, the samples were furnace cooled to ambient temperature. During the brazing, vacuum was maintained at 10^−3^ Pa. The brazed joints with two different coating combinations were designated as BJ-1 and BJ-2, respectively. The brazed specimens were cut with a low-speed diamond saw, hot mounted in a conductive epoxy, and polished with diamond suspensions with particle sizes of 30, 15, 9, and 1 μm, respectively. A JEOL JXA-8530F Plus electron probe microanalyzer (EPMA) combined with a wavelength dispersive spectroscope (WDS) was used for quantitative chemical analyses of selected locations in the brazed zones. Secondary and backscattered electron images (SEI and BEI) of the joints were collected. The operation voltage was kept at 15 kV, and the minimum spot size was 1 μm.

For evaluation of the gas tightness of two brazed joints, the pressure drop method was adopted to measure the in-situ pressure changes inside the brazed samples [29,30,31]. Crofer alloy and YSZ with dimensions of 14 × 14 and 20 × 20 mm^2^ were utilized as the substrates. A hole with a diameter of 8 mm was drilled in the center area of the Crofer alloy in advance and connected to a helium reservoir by a metal tube. BAg-8 filler foil was cut into a hollow preform with an inner area of 8 × 8 mm^2^ and an outer area of 16 × 16 mm^2^ before brazing. After brazing, a SUS310 stainless tube was laser-welded to the drilled Crofer substrate of the brazed sample (The assembly of brazed samples was illustrated in Appendix A), which was connected to a helium reservoir with a positive pressure of 2 psig (pounds per square inch in gauge pressure). The inner pressure of the brazed sample was recorded over time to examine the gas leakage of the brazed joint (The apparatus of pressure drop test was illustrated in Appendix A). Tests were carried out at 25 and 600 °C for 24 h, respectively. After completion of the pressure drop tests, the leak rates of two brazed joints were calculated.

## 3. Results and Discussion

### 3.1. Wetting of the BAg-8 Filler on Metallized Crofer and YSZ Substrates

Figure 2 shows top and transverse views of the wetting test specimens brazed at 850 °C for 600 s. The BAg-8 filler ball well wetted the two metallized substrates. The Crofer substrate coated with a 5-μm Ag layer demonstrated better wetting and spreading characteristics than those of Crofer with a 1.5-μm Ag. The wetting angle of Crofer with a 1.5-μm Ag was 30 degrees, and was decreased to approximately 0 degrees for the Crofer with 5-μm Ag. For the Ag/Cu/Ti-coated YSZ substrate, strong interfacial reactions occurred among the BAg-8 filler, Ag/Cu/Ti metallized layers, and YSZ substrate. The wetting angles in both cases were close to 0 degrees due to reactive wetting. The chemical composition of the Bag-8 filler deviated from the eutectic, and an unmelted residue skull was left on top of the specimen (Figure 2e,g). According to the test results, the Bag-8 filler demonstrated good wetting ability on the metallized Crofer and YSZ substrates, and wetting of the Bag-8 filler ball was more favorable if thicker metallic layers were coated on the two substrates.

### 3.2. YSZ/Ti(0.5μ)/Cu(1μ)/Ag(1.5μ)/Bag-8(50μ)/Ag(1.5μ)/Crofer Joint

Figure 3 shows EPMA BEIs of the BJ-1 vacuum brazed at 850 °C for 600 s, and WDS quantitative chemical analyses of selected locations in Figure 3b–d are listed in Table 2. According to the quantitative analyses, the chemical compositions of the A_1_ and B_1_ positions were close to the original compositions of the YSZ and Crofer substrates. Figure 3b displays a higher magnification of area I in Figure 3a. All metallized layers dissolved into the brazed melt, and the brazed zone mainly consisted of Ag-Cu eutectic (marked C_1_) after brazing. The thicknesses of metallized Ti (0.5 μm), Cu (1 μm), and Ag (1.5 μm) layers were much smaller than that of the BAg-8 foil (50 μm). Therefore, the deviation of the Ag-Cu eutectic was not obvious.

Figure 3c shows a higher magnification of area II in Figure 3b, which was on the YSZ side of BJ-1. The BEI indicated that there was no continuous interfacial reaction layer between the YSZ and braze alloy. A black irregular globular phase (marked D_1_) and dispersed fine particles (marked E_1_) were observed on the YSZ side. According to Table 2, the stoichiometric ratio between Fe and Ti at the D_1_ position was close to 2. Based on the Fe-Ti binary alloy phase diagram, the Fe_2_Ti intermetallic was a non-stoichiometric compound, and the stoichiometric ratio between Fe and Ti may have deviated from 2 [32]. The Fe_2_Ti intermetallic compound was thus confirmed. Position E_1_ in Figure 3c had a chemical composition of Ti and O. However, because these dispersed fine particles were much smaller than the lateral resolution of the spot size (1 μm) in the WDS chemical analysis, they could not be accurately identified.

Figure 4 shows the EPMA WDS element mappings of the YSZ side displayed in Figure 3c. According to Figure 4c,d,f, the Ti readily reacted with Cr and Fe, and the Cr dissolved into the Fe_2_Ti intermetallic (D_1_ position). This result is consistent with the aforementioned result. In addition, Figure 4f shows Ti at the interface between YSZ and the braze alloy. It was deduced that the metallized Ti at the interface contributed to the reactive wetting of the braze melt on the YSZ substrate.

Figure 3d shows a higher magnification of area III in Figure 3b, which was on the Crofer side of BJ-1. The original coated Ag layer on the Crofer substrate completely dissolved into the braze melt, and an interfacial layer formed between the BAg-8 braze and Crofer substrate. The results for positions F_1_, G_1_, and H_1_ in Figure 3d and Table 2 suggest that the interfacial phase was alloyed with elements of Ag, Cu, Cr, Fe, and Ti. The interfacial layer could not be identified due to the limited lateral resolution of WDS analysis. Figure 5 shows EPMA WDS element mappings of the Crofer side in Figure 3d. It is important to note the obvious presence of Ti at the interface of the Crofer side in Figure 5f. The metallized Ti layer on the YSZ substrate dissolved into the braze melt and preferentially reacted with Fe and Cr from the Crofer substrate, as demonstrated in Figure 5c,d,f.

### 3.3. YSZ/Ti(1μ)/Cu(3μ)/Ag(5μ)/BAg-8(50μ)/Ag(5μ)/Crofer Joint

Figure 6a shows EPMA BEIs of BJ-2 after vacuum brazing at 850 °C for 600 s. The YSZ substrate was sequentially coated with layers of Ti (1 μm), Cu (3 μm), and Ag (5 μm) before brazing (Figure 1i–l). WDS quantitative chemical analysis results of selected locations in Figure 6b–d are included in Table 3. Based on Figure 6b and Table 3, locations A_2_ and B_2_ were YSZ and Crofer substrates, respectively. Similar to the results observed in the BJ-1 joint, all the metallized layers, namely, Ag, Cu, and Ti, disappeared from the interface between the braze alloy and YSZ. Dissolution of the coated 5-μm Ag layers into the braze melt on both the YSZ and Crofer substrates resulted in deviation of the eutectic composition of the original BAg-8 braze alloy. Therefore, a primary Ag-rich dendrite, marked D_2_, was observed in the brazed zone. The brazed zone mainly consisted of primary Ag-rich dendrites and Ag-Cu eutectic (location C_2_ in Figure 6b). A few Cu-rich phases in the brazed zone are marked E_2_ in Figure 6b.

Figure 6c shows a higher magnification of area II in Figure 6b, which was on the YSZ side of BJ-2. As on the YSZ side of BJ-1 (Figure 3c), a Fe_2_Ti intermetallic particle, marked F_2_, was observed next to the YSZ substrate. The metallized Ti layer on the YSZ substrate preferentially reacted with Fe from the Crofer substrate. In contrast, low Ti concentrations in the Ag-Cu eutectic, Ag-rich, and Cu-rich phases (locations C_2_–E_2_ in Figure 6b) are listed in Table 3. Figure 6c shows part of the YSZ substrate in contact with the Ag-Cu eutectic, as well as a thin interfacial discontinuous layer, marked G_2_ in Figure 6c. However, because the thickness of the interfacial layer was below 1 μm, quantitative chemical analysis of the phase could not be performed accurately due to the limited lateral resolution of the spot size. As the interfacial G_2_ phase was primarily alloyed with Fe, O, and Ti, it was concluded to be the reaction layer between the braze alloy and YSZ substrate.

Figure 7 shows EPMA WDS element mappings of the YSZ side of the joint in Figure 6c. The active element, Ti, reacted with Fe, Cr, and O to form an interfacial reaction layer, as shown in Figure 7c–f. The thicknesses of Cr and Fe in the reaction layer were less than that of Ti (Figure 7c,d,f). Additionally, different oxygen concentrations were observed at the interface between the braze alloy and YSZ substrate, as shown in Figure 7e. The interfacial reaction layer on the YSZ side of BJ-2 (Figure 6c and Figure 7f) was thicker than that of BJ-1 (Figure 3c and Figure 4f) due to the thicker metallized Ti layer on the YSZ side of BJ-2.

Figure 6d displays a higher magnification of area III in Figure 6b, which was on the Crofer side of BJ-2. According to Figure 6d, the chemical composition of location I_2_ was close to that of the Crofer substrate alloyed with 10.4 at% Ti. According to the Fe-Ti phase diagram, Ti can be alloyed in α-Fe up to 10 at%, which is consistent with the experimental observation [32]. A fine precipitate, marked H_2_ in Figure 6d, had a stoichiometric ratio of Fe to Ti of close to 2. Figure 8 shows EPMA WDS element mappings of the Crofer side of BJ-2 (Figure 6d). The Ti layer on the YSZ side was dissolved into the braze melt at the initial stage of brazing. The Ti in the braze melt diffused towards the Crofer side driven by the concentration gradient during the subsequent brazing. According to Figure 8d,f, the Ti preferentially reacted with Fe from the Crofer substrate.
$$\frac{\left(0.5\times 10^{-4}\times A_{1}\times 4.51\right)}{\begin{array}{c} \left[\left(0.5\times 10^{-4}\times A_{1}\times 4.51\right)+\left(1.0\times 10^{-4}\times A_{1}\times 8.96\right)+\left(1.5\times 10^{-4}\times A_{1}\times 10.49\right)+(50\times 10^{-4}\times A_{2}\times 10.0\right)+\\ \left(1.5\times 10^{-4}\times A_{3}\times 10.49\right)] \end{array}}=0.8\;\mathrm{wt}\%\;\left(\mathrm{for}\;\mathrm{BJ}-1\right).$$
$$\frac{\left(1.0\times 10^{-4}\times A_{1}\times 4.51\right)}{\begin{array}{c} \left[\left(1.0\times 10^{-4}\times A_{1}\times 4.51\right)+\left(3.0\times 10^{-4}\times A_{1}\times 8.96\right)+\left(5.0\times 10^{-4}\times A_{1}\times 10.49\right)+(50\times 10^{-4}\times A_{2}\times 10.0\right)+\\ \left(5.0\times 10^{-4}\times A_{3}\times 10.49\right)] \end{array}}=1.3\;\mathrm{wt}\%\;\left(\mathrm{for}\;\mathrm{BJ}-2\right).$$

The presence of Ti plays a crucial role in the reactive wetting of the YSZ substrate [22]. The Ti concentration in the traditional active braze alloy, Ticusil^®^ (Morgan Advanced Materials, Hayward, CA, USA), is 4.5 wt%. The above equations represent average concentrations of Ti in the two brazed joints, where the density of Ti is 4.51 g/cm^3^, that of Cu is 8.96 g/cm^3^, that of Ag is 10.49 g/cm^3^, and that of BAg-8 is 10.0 g/cm^3^. A_1_ is the area of the YSZ substrate, 2 × 2 cm^2^; A_2_ is the area of the BAg-8 hollow preform, (1.6^2^ − 0.8^2^) cm^2^; and A_3_ is the area of the drilled Crofer substrate, (1.4^2^ – π × 0.8^2^/4) cm^2^. According to the calculation illustrated above, the average Ti concentrations in the braze melts of the BJ-1 and BJ-2 joints were 0.8 wt% and 1.3 wt%, respectively. These values are much smaller than those of active braze alloys, such as Cusil-ABA^®^ (63% Ag-35.25% Cu-1.75% Ti, in wt%) and Ticusil^®^ (68.8% Ag-26.7% Cu-4.5% Ti, in wt%). Therefore, they indicate that decreasing the Ti content in the brazed zone reduced the interfacial reaction between the braze alloy and Crofer substrate, as illustrated in Figure 3d or Figure 6d. Additionally, the reactive wetting between BAg-8 and the metallized YSZ substrate was achieved even when the average Ti concentration was reduced, leading to the formation of crack-free joints.

Figure 9 shows EPMA BEI cross sections of the YSZ side before and after brazing. As shown in Figure 9a, the thickness ratio of Ag/Cu in the metallized YSZ substrate was approximately 1.6. According to the Ag-Cu binary alloy phase diagram, the atomic ratio of Ag/Cu in the eutectic is 1.5 [32]. The chemical compositions of the deposited Ag and Cu layers on the metallized YSZ were very close to that of the Ag-Cu eutectic. Figure 9b shows a fine eutectic, which resulted from the metallized Ag/Cu layers after brazing and was quite different from the coarse eutectic caused by solidification of the BAg-8 filler after brazing. An interface between the two eutectics is indicated by white arrows in Figure 9b. The width of the fine eutectic on the YSZ side of BJ-2 was approximately 7 μm (Figure 9b), which was a little smaller than the width of 8 μm (Figure 9a) of metallized Ag and Cu layers before brazing. Figure 9 also shows that most of the metallized Ti layer with a 1-μm thickness reacted with Fe and formed globular Fe_2_Ti intermetallic compounds. The interfacial reaction layer, which was rich in Ti, was very thin and discontinuous as displayed by white arrows in Figure 9b. Additionally, the BJ-2 joint was free of cracks, and this absence was beneficial to its gas-tight performance.

### 3.4. Pressure Drops Tests and Microstructures of Two Brazed Joints after Tests

Figure 10 shows pressure drops in the leak tests of two joints, BJ-1 and BJ-2, tested at 25 °C under 2 psig helium for 24 h. According to the test results, nearly no drop in pressure occurred; both BJ-1 and BJ-2 demonstrated excellent gas-tight performance at room temperature with smaller amounts of Ti (0.8 and 1.3 wt%) in the braze melt. The improved reactive wetting of the BAg-8 braze alloy on the metallized YSZ substrate and the crack-free interfaces were responsible for the achievement of such gas-tight joints.

Figure 11 shows the pressure drops of BJ-1 and BJ-2 joints, tested at 600 °C for 24 h. As shown in Figure 11a, the inner pressure of the BJ-1 joint decreased gradually from 2.01 to 1.91 psig. The pressure drop of BJ-1 initiated at the beginning of the test. It may have resulted from the weak bonding between the YSZ and BAg-8 filler, as illustrated in Figure 3c. To investigate the amount of gas leakage in the pressure drop test, the leak rate equation below was applied [33,34]:(1)Leak rate=ΔPΔt ×V (mbar×Ls)
where ΔP/Δt is the pressure drop rate, and V is the volume of helium reservoir (10 L). As the inner pressure drop of the BJ-1 joint exhibited a linear decay, the average leak rate was calculated based on the slope of the curve (2.02 × 10^−3^ psig/h) to be 3.88 × 10^−4^ mbar L/s. For the BJ-2 joint (Figure 11b), no helium leakage occurred during 600 °C isothermal holding due to the greater interfacial reaction between YSZ and the braze melt. However, the pressure slightly decreased from 2.02 to 1.98 psig during the cooling cycle of the test. The slope of the cooling stage was approximately 1.78 × 10^−3^ psig/h, and the average leak rate in the cooling cycle of the pressure drop test was equal to 3.40 × 10^−4^ mbar L/s. The BJ-2 joint demonstrated slightly better gas-tightness and thermal stability at 600 °C due to the greater interfacial reaction between the YSZ and BAg-8 filler in the joint.

## 4. Conclusions

Dissimilar vacuum brazing of metallized YSZ and Crofer was performed with the BAg-8 braze alloy. The YSZ substrate was metallized by sequentially sputtering Ti (0.5/1 μm), Cu (1/3 μm), and Ag (1.5/5 μm) layers, and the Crofer substrate was coated with Ag (1.5/5 μm), respectively; both joints were free of cracks. The important conclusions are:

(1)The BAg-8 filler well wets both metallized YSZ and Crofer substrates. The wetting angle of Crofer with 1.5-μm Ag was 30 degrees, and it was decreased to approximately 0 degrees for the Crofer with 5-μm Ag. For the Ag/Cu/Ti-coated YSZ substrate, strong interfacial reactions occurred among the Bag-8 filler, Ag/Cu/Ti metallized layers, and YSZ substrate. The wetting angles in both cases were close to 0 degrees due to reactive wetting;(2)The brazed zone primarily consists of Ag-Cu eutectic. The metallized Ti layer dissolves into the braze melt, and the Ti preferentially reacts with YSZ and Fe from the Crofer substrate. Globular Fe_2_Ti intermetallic compounds form on the YSZ side of the joint. The interfacial reaction among Ti and the two substrates increases when the thickness of the metallized Ti layer is increased from 0.5 to 1 μm;(3)No pressure drop was detected for the two BAg-8 brazed metallized YSZ and Crofer joints during testing at room temperature for 24 h. For the YSZ/Ti(0.5μ)/Cu(1μ)/Ag(1.5μ)/BAg-8(50μ)/Ag(1.5μ)/Crofer joint tested at 600 °C for 24 h, the pressure of helium gradually decreased from 2.01 to 1.91 psig. In contrast, the helium pressure drop test of YSZ/Ti(1μ)/Cu(3μ)/Ag(5μ)/BAg-8(50μ)/Ag(5μ)/Crofer joint showed a slight decrease from 2.02 to 1.98 psig during the cooling cycle of the test. The greater interfacial reaction between the metallized YSZ and BAg-8 filler due to the thicker metallized Ti layer on the YSZ substrate was responsible for the improved gas-tight performance of the joint;(4)The combination of the BAg-8 filler foil and metallized YSZ and Crofer substrates demonstrates a potential approach to create joints for gas-tight applications.

## Figures and Tables

**Figure 1 materials-15-00939-f001:**
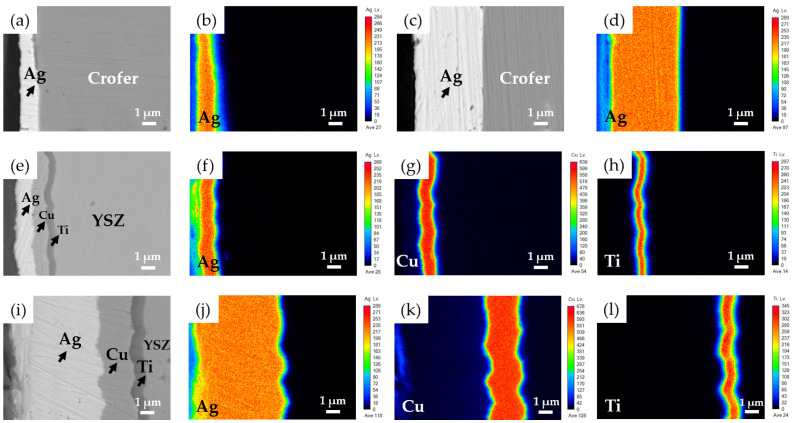
Cross-sectional BEIs and element mappings of the Crofer and YSZ substrates after sputtering: (**a**) Ag(1.5μ)/Crofer, (**b**) Ag mapping in (**a**), (**c**) Ag(5μ)/Crofer, (**d**) Ag mapping in (**c**), (**e**) Ag(1.5μ)/Cu(1μ)/Ti(0.5μ)/YSZ, (**f**–**h**) Ag, Cu, and Ti mappings in (**e**), (**i**) Ag(5μ)/Cu(3μ)/ Ti(1μ)/YSZ, and (**j**–**l**) Ag, Cu, and Ti mappings in (**i**).

**Figure 2 materials-15-00939-f002:**
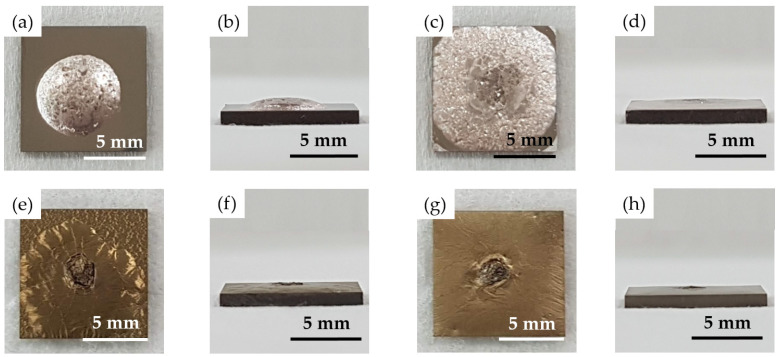
Top and transverse views after wetting tests at 850 °C/600 s for the Bag-8 filler ball on (**a**,**b**) Ag(1.5μ)/Crofer, (**c**,**d**) Ag(5μ)/Crofer, (**e**,**f**) Ag(1.5μ)/Cu(1μ)/Ti(0.5μ)/YSZ, and (**g**,**h**) Ag(5μ)/Cu(3μ)/Ti(1μ)/YSZ.

**Figure 3 materials-15-00939-f003:**
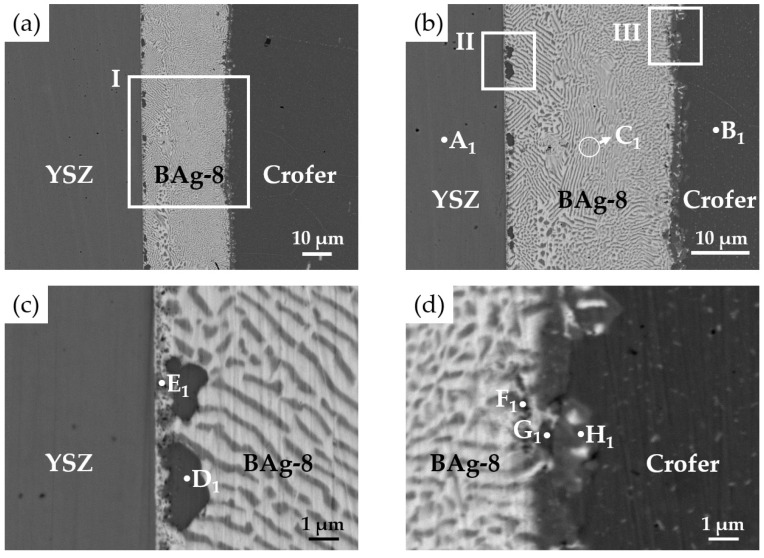
EPMA BEIs of the BJ-1 vacuum brazed at 850 °C for 600 s: (**a**) Cross section of the joint, (**b**) higher magnification of area I in (**a**), (**c**) YSZ side of area II in (**b**), and (**d**) Crofer side of area III in (**b**).

**Figure 4 materials-15-00939-f004:**
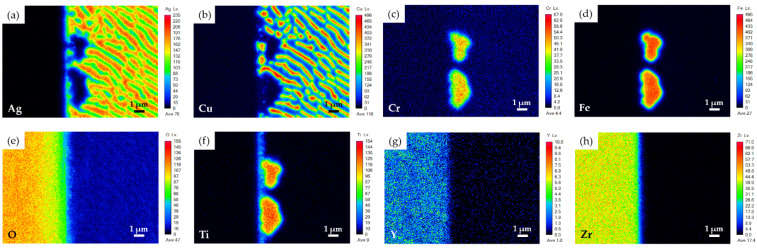
EPMA WDS element mappings of YSZ side in Figure 3c: (**a**) Ag, (**b**) Cu, (**c**) Cr, (**d**) Fe, (**e**) O, (**f**) Ti, (**g**) Y, and (**h**) Zr.

**Figure 5 materials-15-00939-f005:**
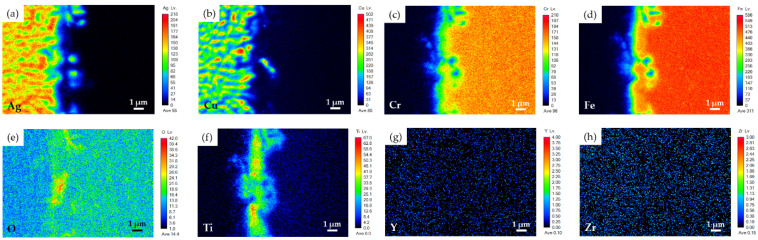
EPMA WDS element mappings of Crofer side in Figure 3d: (**a**) Ag, (**b**) Cu, (**c**) Cr, (**d**) Fe, (**e**) O, (**f**) Ti, (**g**) Y, and (**h**) Zr.

**Figure 6 materials-15-00939-f006:**
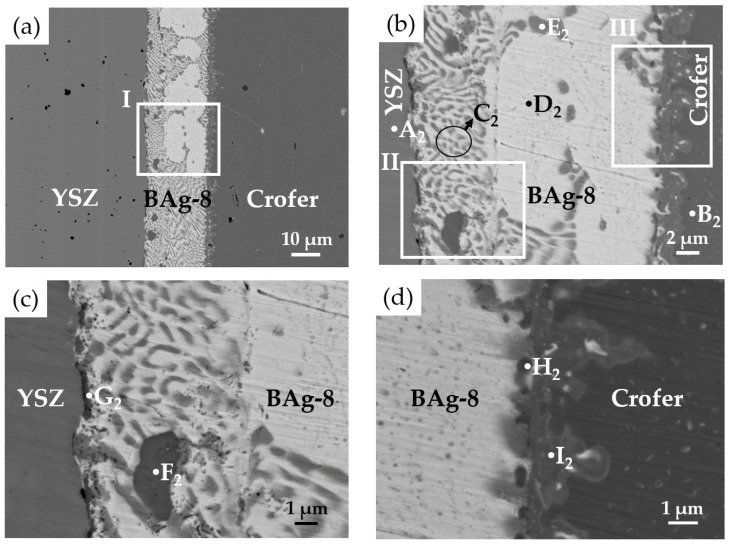
EPMA BEIs of the BJ-2 vacuum brazed at 850 °C for 600 s: (**a**) Cross section of the joint, (**b**) higher magnification of area I in (**a**), (**c**) YSZ side of area II in (**b**), and (**d**) Crofer side of area III in (**b**).

**Figure 7 materials-15-00939-f007:**
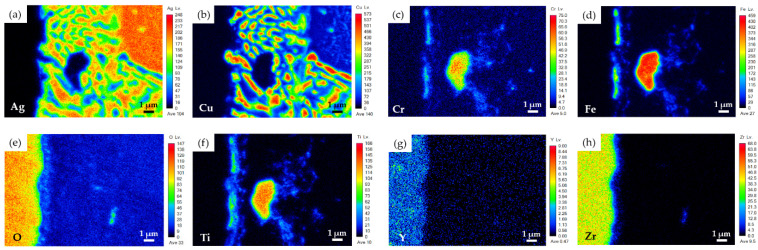
EPMA WDS element mappings of YSZ side in Figure 6c: (**a**) Ag, (**b**) Cu, (**c**) Cr, (**d**) Fe, (**e**) O, (**f**) Ti, (**g**) Y, and (**h**) Zr.

**Figure 8 materials-15-00939-f008:**
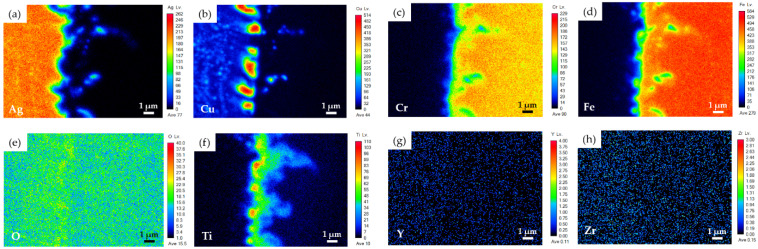
EPMA WDS element mappings of the Crofer side in Figure 6d: (**a**) Ag, (**b**) Cu, (**c**) Cr, (**d**) Fe, (**e**) O, (**f**) Ti, (**g**) Y, and (**h**) Zr.

**Figure 9 materials-15-00939-f009:**
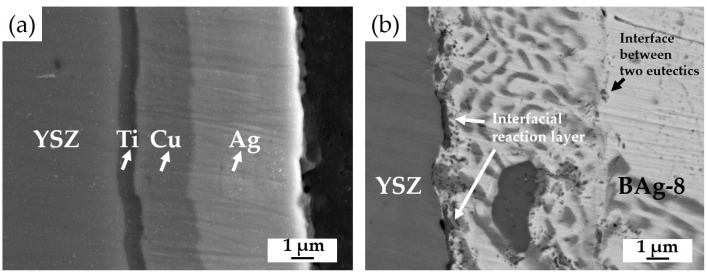
EPMA BEI cross sections: (**a**) Ag(5μ)/Cu(3μ)/Ti(1μ)/YSZ substrate before brazing and (**b**) the YSZ side of BJ-2 after brazing.

**Figure 10 materials-15-00939-f010:**
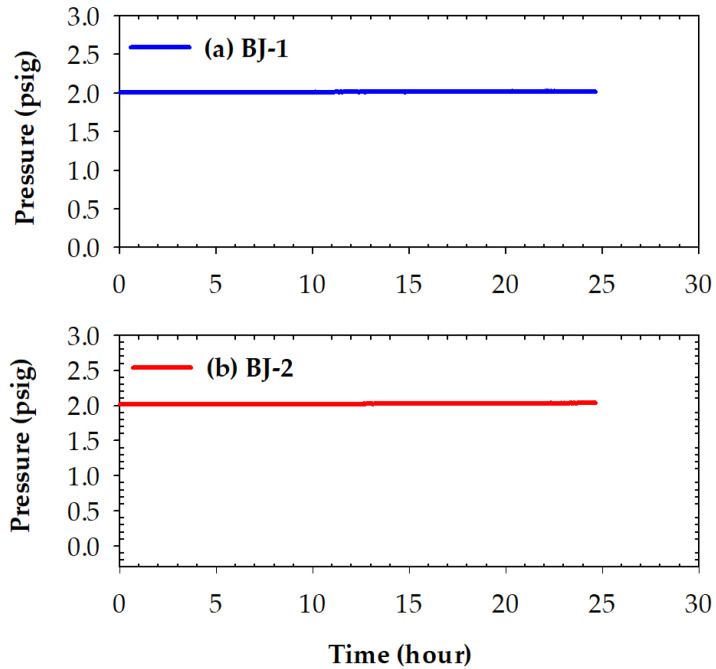
Pressure drops of two joints tested at 25 °C for 24 h: (**a**) BJ-1 and (**b**) BJ-2.

**Figure 11 materials-15-00939-f011:**
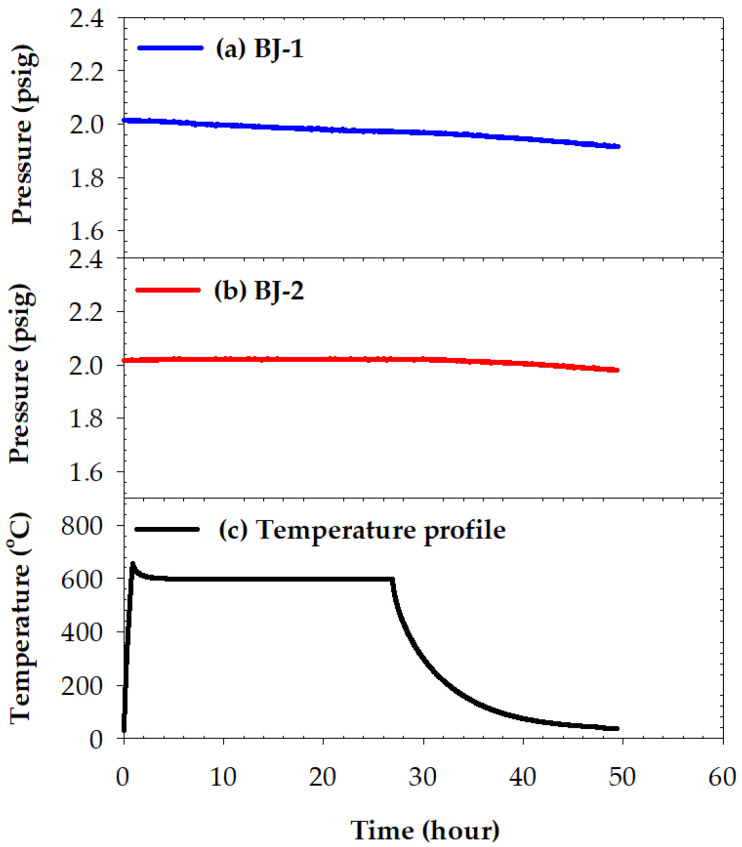
Pressure drops in leak tests of two joints tested at 600 °C for 24 h: (**a**) BJ-1, (**b**) BJ-2, and (**c**) temperature profile in the test.

**Table 1 materials-15-00939-t001:** Sputtering parameters and thicknesses of coating layer(s) on YSZ and Crofer substrates.

Substrate	Sputtering Power (600 s) ^1^	Thickness (μm) ^1^
YSZ-1	Ti: 175 W, Cu: 57 W, Ag: 65 W	Ti: 0.5, Cu: 1.0, Ag: 1.5
YSZ-2	Ti: 357 W, Cu: 172 W, Ag: 217 W	Ti: 1.0, Cu: 3.0, Ag: 5.0
Crofer-1	Ag: 65 W	Ag: 1.5
Crofer-2	Ag: 217 W	Ag: 5.0

^1^ The method to measure coating thickness and correlation curve between thickness and sputtering power are illustrated in Appendix A, respectively.

**Table 2 materials-15-00939-t002:** EPMA WDS quantitative chemical analyses of BJ-1 in Figure 3b–d.

Element ^1^/at%	Ag	Cu	Cr	Fe	O	Ti	Y	Zr	Phase/Alloy
A_1_	0.0	0.2	0.0	0.0	61.1	0.0	2.2	36.4	YSZ ^3^ (substrate)
B_1_ ^1^	0.1	0.1	24.9	73.0	0.0	0.2	0.0	0.0	Crofer ^4^ (substrate)
C_1_ ^5^	72.0	27.8	0.0	0.1	0.0	0.0	0.0	0.0	Ag-Cu eutectic ^5^
D_1_	1.1	1.1	7.1	58.3	0.3	31.7	0.0	0.1	Fe_2_Ti
E_1_ ^2^	10.0	60.7	1.1	8.2	5.5	11.5	0.1	2.5	---
F_1_ ^2^	27.7	49.3	4.8	11.1	0.4	6.4	0.0	0.0	---
G_1_ ^2^	10.5	2.3	16.9	54.0	5.8	9.0	0.0	0.0	---
H_1_ ^1^	5.3	36.4	13.9	37.9	0.0	5.0	0.0	0.0	---

^1^ Others: Si, Nb, W, Ni, Mn, and La (every element < 1.0 at%) are not shown in the table; ^2^ areas of E_1_–G_1_ are less than the spot size (1 μm) used in the EPMA WDS analyses; ^3^ nominal composition of YSZ is 66.0 at% O, 2.0 at% Y, and 32.0 at% Zr (trace elements in ppm level); ^4^ nominal composition of Crofer is 23.5 at% Cr and 73.5 at% Fe (Si, Nb, W, Ni, Mn, and La for balance); ^5^ the spot size of chemical analyses at position C_1_ was 3 μm.

**Table 3 materials-15-00939-t003:** EPMA WDS quantitative chemical analyses of BJ-2 in Figure 6b–d.

Element ^1^ /at%	Ag	Cu	Cr	Fe	O	Ti	Y	Zr	Phase/Alloy
A_2_	0.0	0.2	0.0	0.1	63.5	0.0	1.5	34.6	YSZ ^3^ (substrate)
B_2_ ^1^	0.1	0.1	25.0	73.0	0.1	0.3	0.0	0.0	Crofer ^4^ (substrate)
C_2_ ^5^	71.6	28.2	0.0	0.0	0.0	0.0	0.0	0.0	Ag-Cu eutectic ^5^
D_2_	90.7	8.4	0.1	0.2	0.3	0.0	0.0	0.0	Primary Ag
E_2_	3.2	94.7	0.2	0.9	0.1	0.8	0.0	0.0	Cu-rich
F_2_	0.9	1.3	9.0	56.4	0.2	31.5	0.0	0.2	Fe_2_Ti
G_2_ ^2^	13.4	1.9	3.9	33.6	10.1	20.1	0.7	15.6	---
H_2_ ^2^	14.4	4.2	5.8	47.8	0.5	26.6	0.0	0.0	Fe_2_Ti
I_2_ ^1^	0.5	0.8	24.0	62.2	0.1	10.4	0.0	0.0	Crofer alloyed with Ti

^1^ Others: Si, Nb, W, Ni, Mn, and La (every element < 1.0 at%) are not shown in the table; ^2^ areas of G_2_ and H_2_ are less than the spot size (1 μm) used in the EPMA WDS analyses; ^3^ nominal composition of YSZ is 66.0 at% O, 2.0 at% Y, and 32.0 at% Zr (Trace elements in ppm level); ^4^ nominal composition of Crofer is 23.5 at% Cr, and 73.5 at% Fe (Si, Nb, W, Ni, Mn, and La for balance); and ^5^ the spot size of chemical analyses at position C_2_ was 3 μm.

## Data Availability

Not applicable.

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
