# Peer review of "Vacuum Brazing of Metallized YSZ and Crofer Alloy Using 72Ag-28Cu Filler Foil"

_materials, 2022, doi:10.3390/ma15030939_

Round 1

Reviewer 1 Report

The authors have prepared an interesting manuscript on vacuum brazing of metallized YSZ and Crofer alloy using a silver-based filler metals. However, minor modifications are suggested below to improve the quality of the manuscript:

-              Abstract: The interfacial reaction of Ti and the two… (confusing with the word “and”)

-              Non-uniform term was used for Bag-8 filler metal. Please use the standard terms.

-              Materials and Experimental Procedures: For the last paragraph, it is suggested to aid the explanation with illustration (with scale and labels).

-              Improve the font and label colour for Figure 3 and 6. The current font and label for the figures mostly are not clear for printing in black and white.

-              Page 8, last paragraph: The interfacial reaction layer, which was rich in Ti, was very thin and discontinuous – Please mark it on the Figure.

-              Figure 11: The x-axis for the time (hours), is it correct?

Reviewer 2 Report

The manuscript reports an interesting work on the vacuum brazed joint of metallized YSZ and Crofer alloy using BAg-8 filler foil. In general, the work is relevant and interesting, but a major revision is required before publication, as detailed below:

  1. A schematic view of the pressure drop method and assemblies of the samples for this test is required either in the manuscript or supplementary information.
  2. The text of the work reports that during preliminary metallization, the thickness of the copper layer on the YSZ substrate is 1 and 3 µm. However, in Fig. 1, both on a cross-section (Fig. 1i) and on the map of the distribution of elements (Fig. 1k), it can be seen that the thickness of the copper layer is about 2 μm. Accordingly, if we take into account that the scale bars in fig.1 for the thicknesses of the remaining layers gives the correct values, then it makes sense to change the layer thickness to 2 microns in the text of the article.
  3. And the next question, which arises due to the discrepancy between the copper layer and the specified thickness, is that how to justify the increase in power from 57 to 172 W when this layer was sputtered.

  1. In Section 3.1: Please add quantification values ​​for wettability, eg contact angle of BAg-8 filler. And, accordingly, add these results to the conclusions section, if possible.

  1. Tables 2, 3. The authors do not give the initial compositions of the YSZ substrates and the Crofer alloy, therefore it is difficult to assess whether the diffusion of the elements of metallized coatings and the BAg-8 filler foil took place and to what depth.

  1. Tables 2, 3. The chemical composition of the eutectic is indicated incorrectly, since the eutectic is two-phase, and the place of analysis is indicated by point C1, it is not clear which of the two phases has been determined. The authors should use an area instead of the point analysis.

  1. There is no Figs. 4 and 7 in the manuscript. Please check and correct the numbering of the figures.

  1. Why there are high concentrations of Ti at the Crofer side of brazing even higher than that at the YSZ side, given that Ti was coated on the YSZ? Is it possible that high amount of Ti diffuse from YSZ to Crofer side during 10 min brazing at 850 °C? What is the driving force for diffusion?

  1. Page 8. Before the formulas for calculating the average titanium content in compounds, it is necessary to add several sentences about why this calculation is done. Now the transition from the description of Fig. 6 and 8 in the paragraph above to this calculation is not clear.

  1. General note - figures and tables are provided before they are mentioned in the text. Check, please.

Reviewer 3 Report

Purely technological work, the reason for choosing such and not other materials for metallization has not been explained. Introduction should be more concise. The areas in Fig. 1 (also the remaining ones) should be better marked, e.g. with arrows. There are also different magnifications, eg Fig. 1a, e, and they have different magnifications than others, what is the reason for this? In general, the drawings should be clearer. The conclusions are very obvious. 

Round 2

Reviewer 2 Report

The authors provided reasonable answers and corrections to the manuscript. I recommend publication after applying two more corrections:

1- The procedure for measuring wetting angle need to be added to the materials and method section.

2- The changes to reflect the composition of eutectic phase (comment 5 from previous review) is not satisfying. The authors should provide the composition from a region (instead of point C2 in Fig. 6b) to give a more accurate chemical composition of eutectic phase.
